# Recent Development in Earth-Abundant Kesterite Materials and Their Applications

**Ahmet Sencer Nazligul** [ID]**, Mingqing Wang** [ID] **and Kwang Leong Choy \***

Institute for Materials Discovery, University College London, London WC1E 7JE, UK;
ahmet.nazligul.17@ucl.ac.uk (A.S.N.); mingqing.wang@ucl.ac.uk (M.W.)

**\*** Correspondence: k.choy@ucl.ac.uk

**Abstract:** Kesterite $Cu_2ZnSnS_4$ (CZTS) has attracted attention as an earth-abundant alternative to commercially successful CIGS solar cells. CZTS exhibits decent optoelectrical properties while having excellent stability on top of being an earth-abundant, low-cost and non-toxic material. Therefore, in recent years, there has been a significant research effort to develop CZTS-based devices. The efficiency of CZTS solar cells reached 12.6% in 2013, and this was a remarkable achievement at the time. However, the efficiency of these devices has been stagnant since then while emerging technologies, most notably perovskite solar cells, keep breaking record after record. Currently, CZTS research focuses on discovering the secrets of material properties that hinder the efficiency of CZTS solar cells while branching out to develop alternative applications for this material. In this review, we summarize the interesting properties of CZTS as well as its promising applications, which include thin-film solar cells, charge-transfer layers in perovskite solar cells, and photoelectrochemical water splitting while briefly commenting on its other possible applications.

**Keywords:** kesterite; CZTS; thin-film solar cells; charge-transfer layer; photoelectrochemical water splitting

## 1. Introduction

Parallel to increasing public concerns about environmental issues and climate change, investments and research activities related to renewable energy have grown substantially in the past decade. From the commercial side of the spectrum, the cost of renewable energy applications has been reduced to a level where it can compete with traditional energy sources, while on the research side, new technologies with better performance have been developed. As a result of these efforts, in the past seven years, annual additional renewable energy capacity has increased more rapidly than all non-renewable sources combined, with solar photovoltaic (PV) capacity having the highest increase among the other renewable energy sources [1,2].

At present, the solar PV market is dominated by crystalline silicon (c-Si) solar cells, representing more than 90% of the market share. The record efficiency for c-Si technology is 26.7%, which is close to its theoretical efficiency limit of 29.4%; thus, little improvement can be expected [3,4]. As Si has an indirect bandgap and, consequently, a low absorption coefficient, c-Si devices require a thick layer of absorber material. Furthermore, Si has a low tolerance for impurities and expensive high purity crystals are required, which, taken together with the thick absorber layer, significantly increases costs. Thin-film PV technology is a promising, low-cost alternative to c-Si solar cells. These devices are based on direct bandgap semiconductors with high absorption coefficients, which require relatively thin absorber layers and can also be deposited on light, flexible substrates, making them compatible with roll-to-roll production techniques. As a result, these devices may offer significant cost advantages over conventional silicon cells.

Although, in theory, thin-film devices have many advantages over c-Si solar cells, and a handful of thin-film technologies have already reached efficiencies of over 20% (CdTe (22.1%), CIGS (23.4%), perovskite (25.2%)) [5], only $\approx$ 5% of global PV production consists of thin-film devices [6]. This is the result of several challenges; CdTe consists of toxic elements and CIGS requires rare-earth elements. perovskite solar cells (PSCs), which have rapidly improved in efficiency from 3.8% in 2009 to 22.7% in 2016, are the front runners for new generation PV devices [7,8]; however, they suffer from stability issues which hinder their commercial potential. Ideally, a solar cell technology needs to be efficient, cheap, non-toxic, earth-abundant and stable. At present, there is not a single solar cell that meets all these criteria. To solve these problems and make the ultimate solar cell, non-toxic, earth-abundant and stable materials need to be explored.

One promising group of materials is kesterite $Cu_2ZnSnS_4$ (CZTS) and related materials $Cu_2ZnSnSe_4$ (CZTSe) and $Cu_2ZnSn(S,Se)_4$ (CZTSSe). CZTS has been developed by replacing the scarce elements in CIGS with relatively abundant Zn and Sn. It has similar optoelectronic properties with CIGS, such as showing p-type conductivity while having a high absorption coefficient with a similar bandgap. Therefore, using CZTS as a light absorber has been seen as the natural application for this material. CZTS solar cells have shown a remarkable improvement in performance, reaching 12.6% efficiency in 2013 with a CZTSSe absorber [9]. However, the performance of these devices has not experienced a significant improvement since then despite much effort. This is as a result of several challenges, some of which include an abundance of deep level defect states, a narrow phase stability region and non-ideal device architecture. Currently, many researchers are working on CZTS to solve the issues that hinder the potential of CZTS solar cells. Moreover, there has been an increasing number of papers that use CZTS outside its conventional usage, such as in charge-transfer layers, water splitting, thermoelectric devices and sensors, although the main focus is still on solar cell applications. This review summarizes the interesting properties of CZTS as well as giving brief status updates on its various applications.

## 2. Material Properties

Similar to chalcopyrite-type $CuInS_2$ (CIS), kesterite $Cu_2ZnSnS_4$ - $Cu_2ZnSn(S,Se)_4$ - $Cu_2ZnSnSe_4$ are members of the Cu-chalcogenide material family [10,11]. Both chalcopyrite and kesterite are effectively substituted derivatives of the Zinc blende crystal structure of ZnS. These structures have sulphur and/or selenium anions and a range of metal cations (see Figure 1) [11]. Complex compounds can be derived from the binary Zinc blende structure by substituting one of the components with another element as long as the charge-balance is maintained [12,13]. The binary Zinc blende consists of group II-VI elements in which the cation is $Zn^{2+}$. By substituting Zn with a combination of $Cu^{1+}$ and $In^{3+}$, CIS can be derived, which has the overall structure of I-III-VI$_2$. Quaternary CZTS with a I$_2$-II-IV-VI$_4$ structure can be derived from CIS by replacing $In^{3+}$ with $Zn^{2+}$ and $Sn^{4+}$ [11]. As long as the octet rule is satisfied, it is possible to derive several ternary/quaternary semiconductor materials, including $CuInS_2$, $CuGaS_2$, $CuFeS_2$, $Cu_2ZnSnS_4$, $Cu_2FeSnS_4$, $Cu_2CdSnS_4$, $Cu_2ZnGeS_4$ and their selenide counterparts [14].

It has been shown theoretically and experimentally that tetragonal kesterite is the most stable crystal structure for $Cu_2ZnSn(S,Se)_4$, although several metastable structures including stannite, Zinc blende, wurtzite, wurtzite-kesterite and wurtzite-stannite have also been observed [15,16]. Kesterite CZTS has a direct bandgap of 1.5 eV and a high absorption coefficient ($>10^4$) which is close to the ideal for a single-junction solar cell. The bandgap can also be tuned by altering the S/Se ratio between 1.0 eV for pure selenide and 1.5 eV for pure sulfide compounds [17].

The band structure of CZTS is somewhat similar to other Cu-chalcogenides. Valance band maximum (VBM) of CZTS is determined by the antibonding state of a hybrid *p-d* orbital composed of anion *p* and Cu *d* orbitals [18,19]. Because the orbital level of Se is higher than that of S, incorporation of Se anions result in higher VBM and lower bandgap [20]. On the other hand, conduction band minimum (CBM) is mainly affected by the antibonding state of the hybrid *s-p* orbital formed between the Sn *s* orbital and the anion *p* orbital. Even though the Se *s* level has higher energy than the S *s* level, the CBM of sulfides is higher than that of selenides as a result of a shorter Sn-S bond compared



to the Sn-Se bond. As mentioned previously, incorporation of Se is one way to tune the bandgap. However, bandgap can also be changed by substituting Cu or Sn cations.

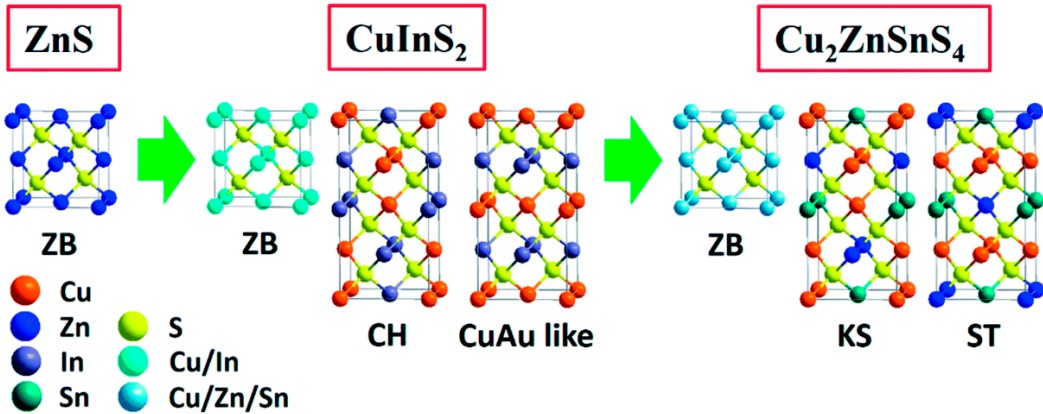

**Figure 1.** Crystal structures of Zinc blende derivatives showing the transformation from ZnS to CuInS$_2$ to Cu$_2$ZnSnS$_4$ (reused from [11] with permission, licensed by RSC).

Although similarities between CIGS and CZTS greatly helped in developing CZTS-based solar cells, there are several differences between these two materials that need to be addressed. One big difference between CZTS and CIGS is the defect chemistry. Similar to CIGS, CZTS is intrinsically a p-type semiconductor as a result of acceptor defects such as copper vacancies (V$_{Cu}$) and copper-on-zinc (Cu$_{Zn}$) anti-site defects having lower formation energies than donor defects [21]. However, the main defect in CZTS is Cu$_{Zn}$, while V$_{Cu}$ is more common in CIGS. Figure 2a shows that Cu$_{Zn}$ has negative formation energy, which means it forms spontaneously. The reason that Cu$_{Zn}$ has such low formation energy is a topic of debate; however, many researchers attribute this to the close atomic radiuses of Cu and Zn (145 pm and 142 pm, respectively). The calculated defect ionisation levels are shown in Figure 2b [22]. Although the existence of shallow defects such as V$_{Cu}$ is quite beneficial to enhance the p-type conductivity [23], deep level defects such as Cu$_{Zn}$ reduce the effective bandgap of the material, causing low open-circuit voltage (V$_{oc}$) in CZTS-based solar cells [24]. A high amount of deep level defects also increases the recombination rate. Photoluminescence bands associated with kesterite materials in Figure 2c are consistently asymmetric and below the bandgap of the material, meaning that the main recombination occurs band-to-impurity rather than band-to-band [24,25]. Gokmen et al. suggested that band tailing occurs in CZTS due to electrostatic potential fluctuations because of defect complexes such as [Cu$^-_{Zn}$ + Zn$^+_{Cu}$] [26]. Another potential problem is the existence of secondary phases. As CZTS is a fairly complex material with many components, its phase stability region is narrow and, depending on the deposition conditions, binary and ternary phases including ZnS, Cu$_x$S, SnS$_x$ and Cu$_2$SnS$_3$ can co-exist with the quaternary Cu$_2$ZnSnS$_4$ phase, which often negatively affects the device performances. One common method to avoid both Cu$_{Zn}$ defects and Cu-rich secondary phases while promoting beneficial V$_{Cu}$ is intrinsic doping, which is achieved by employing Cu-poor and Zn-rich compositions rather than stoichiometric composition [27–31].

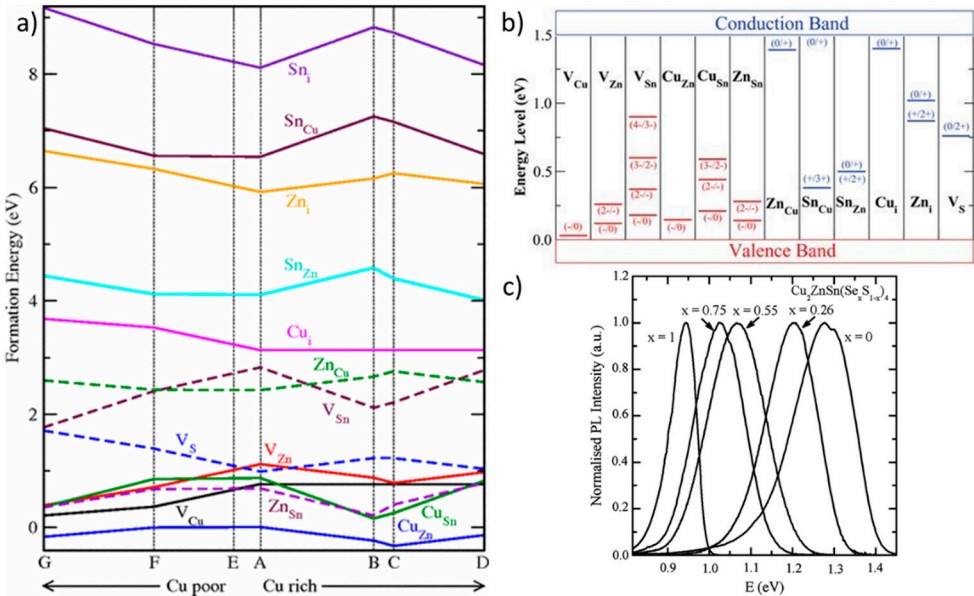

**Figure 2.** (**a**) Formation energies of intrinsic defects in $Cu_2ZnSnS_4$ (CZTS) (reused from [21] with permission, licensed by the American Physical Society), (**b**) Ionisation levels of these defects in the bandgap of CZTS (Red bars for acceptor defects, blue bars for donor defects) (reused from [22] with permission, licensed by John Wiley and Sons), (**c**) Low temperature photoluminescence spectra of $Cu_2ZnSn(S_xSe_{1-x})_4$ with different S/Se ratios (reused from [25] with permission, licensed by Elsevier).

## 3. CZTS Absorber Layers in Thin-Film Solar Cells

The first CZTS solar cell with a demonstrable power conversion efficiency (0.66%) was reported by Katagiri et al. In 1997 [32], before rapidly increasing to 12.6% efficiency by 2013 [9]. Despite the common assumption that vacuum-based methods result in better device performance relative to solution-based methods, since 2009, most record devices, including the current record, have been deposited by solution-based methods [9,33–35]. This is a key advantage for the possibility of mass-production as these methods are relatively cheap and easily scalable. In this review, we have summarized some of the promising solution-based approaches.

Solution based deposition methods involve dissolving metal chalcogenides, oxides or salts in a solvent to prepare a precursor solution. Then, the solution is deposited onto a substrate through the desired method, such as spin-coating, dip-coating and spray pyrolysis. The deposition step is repeated several times to reach a certain thickness, and in-between each step the solvent is removed by generally applying low to medium heat. In the last step, the films are annealed at high temperature to form the quaternary kesterite phase. Generally, the final annealing is done in a sealed box with the presence of excess sulphur or selenium, a process called sulfurization or selenization, to both avoid chalcogen loss and to tune the S/Se ratio. The quality of the final films is highly dependent on the quality of the solutions as well as deposition and annealing conditions. Solutions that contain agglomerations or impurities and precursors with wrong oxidation states can lead to poor quality films, while incomplete solvent removal in between each step in the multistep deposition might cause non-uniformities in the film or prohibit the grain growth during nucleation.

The record efficiency device was deposited by spin-coating a metal chalcogenides-hydrazine solution. The solubility of metal chalcogenides is generally low in most solvents. Mitzi et al. first introduced a 'dimensional reduction' by using hydrazine as the solvent to fabricate CIGSSe devices [36,37]. After demonstrating high efficiencies in CIGSSe cells, this method was modified for CZTSSe cells [9,34,38–40]. In this approach, the addition of hydrazine as a strong ionic reagent breaks the metal–anion framework into discrete metal chalcogenide anions, as shown in Figure 3. The resulting solution can then be effectively coated onto the substrate and transformed into the

desired phase by annealing. The record efficiency CZTSSe device was fabricated by the dissolution of individual $Cu_2S$–S and SnSe–Se–Zn powders in hydrazine to form the precursor slurry which was spin-coated on the substrate and annealed [9].

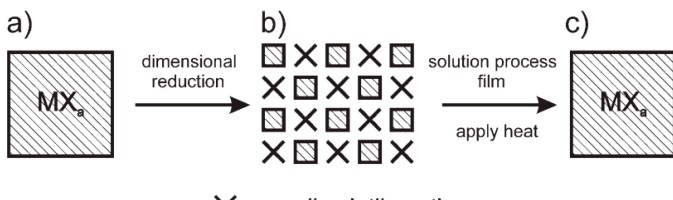

**Figure 3.** Schematic drawing of dimensional reduction process (reused from [37] with permission, licensed by John Wiley and Sons).

Even though devices processed using hydrazine solutions achieve high efficiencies, hydrazine is both explosive and toxic. Therefore, it is difficult to apply this to mass production. As such, finding a safe, environmentally friendly alternative is a priority. Instead of metal chalcogenides, the use of metal chlorides, acetates and nitrates is regarded as a convenient alternative. As these salts are highly soluble in water and several organic solvents, the components of CZTS can be dissolved in a single solution which can be directly coated onto a substrate with similar processing steps to the hydrazine-based approach. Su et al. dissolved metal acetate and chloride salts with thiourea in 2-methoxyethanol to produce CZTS solar cells with efficiencies of 5.1% [41], with the same technique resulting in 8.25% efficiency for CZTSSe cells [42]. Another promising solvent for spin-coating is dimethyl sulfoxide (DMSO). Ki and Hillhouse first fabricated a CZTSSe cell with 4.1% efficiency using this method in 2011 [43], before improving the solution preparation by adding the precursors step-by-step (as seen in Figure 4) to completely reduce $Cu^{2+}$ to $Cu^+$, resulting in an efficiency of 8.32% [44]. Finally, Xin et al. reached 11.8% efficiency with this method by optimizing the selenization process and applying Li doping [45]. Several other groups using this method also achieved efficiencies above 10%, showing the repeatability of the process [46,47]. All previously mentioned CZTS-based solar cells were deposited on top of Mo-coated soda lime glass (SLG) substrates. Instead of SLG, Calvet et al. developed a novel solar cell device that employed ceramic substrates. Cells were prepared by doctor blading and reached 2% efficiency [48].

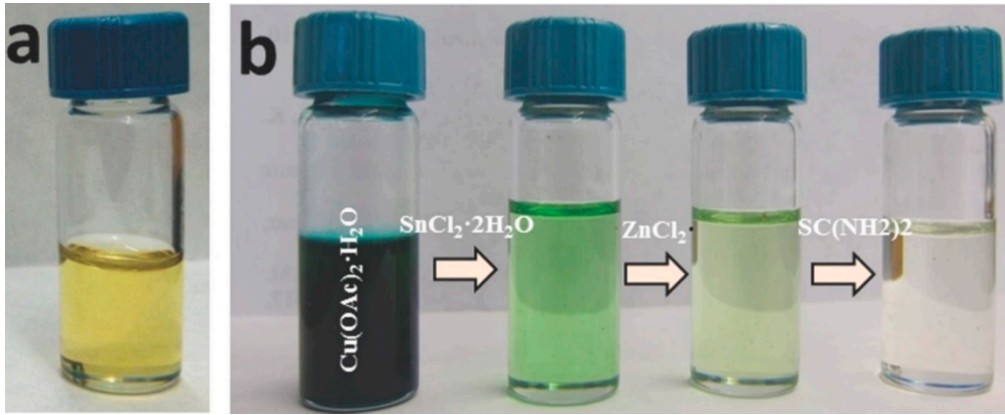

**Figure 4.** Redox equilibrium can be achieved by adding precursors in a specific order. Adding precursors simultaneously usually results in a light-yellow solution while step by step addition results in a colourless solution (reused from [44] with permission, licensed by John Wiley and Sons).

Several strategies are currently under investigation to find ways to overcome the various challenges of high efficiency CZTS solar cells. Like other chalcogenides, doping and substitution can be powerful tools to adjust various material properties. A list of interesting dopants and substitutes is given in

Figure 5 [49]. Briefly, CZTS can be extrinsically doped by introducing alkali metals such as Na, K and Li, while components of CZTS can be substituted by other elements with the same oxidation state.

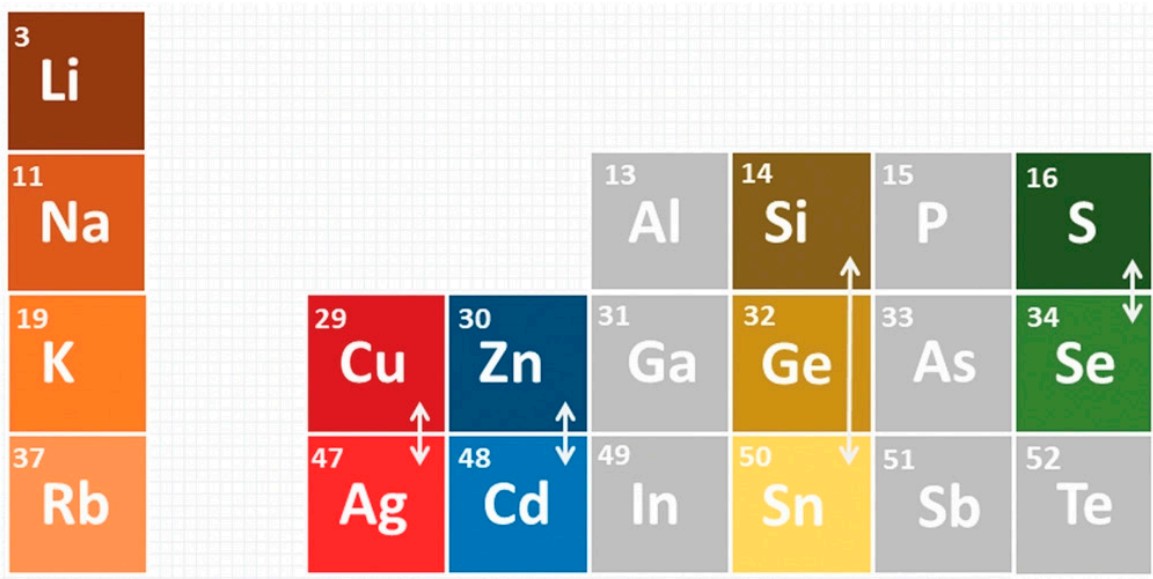

**Figure 5.** Close substitutes for components of $Cu_2ZnSnS_4$ represented in a reduced periodic table (reused from [49] with permission, licensed by John Wiley and Sons).

Alkali metal doping has been adapted for CZTS since the early days of development because it has been proven to be highly beneficial for its predecessor CIGS solar cells [50]. Sodium (Na) is a popular dopant for CZTS, and it has been observed to have a positive influence on grain size and charge-carrier properties [51,52]. Gershon et al. suggested that the ternary Na-Zn-S liquid phase, which is stable at temperatures commonly used for the heat-treatment of CZTS devices, unpins the Zn-rich secondary phases such as ZnS from grain boundaries and accumulates them at the back or surface of the film, allowing larger grains to grow [51]. In addition to improving the microstructure, Na doping also passivates non-radiative defects and increases the hole concentration [52–54]. Xin et al. studied the effects of lithium (Li) doping and found that Li doping helps to repel minority charge-carriers from grain boundaries (GBs) by inverting the polarity of the electric field at the GBs [45]. It is also reported that Li doping increases the bandgap and grain sizes [55]. In our group, Altamura et al. studied the effects of alkali dopants Na, Li and Rb [56]. CZTS films were fabricated through electrostatic spray-assisted vapour deposition, and doping was achieved by simply soaking as-deposited films in water solutions that contained the alkali dopants. The highest efficiency of 5.7% was achieved with Rb doping, followed by 5.5% with Na doping and 4.8% with Li doping. Lastly, potassium (K) doping improved (112) the preferred orientation in the CZTS crystal lattice while increasing the grain sizes and reducing ZnS secondary phases [57].

Apart from doping, cation substitution is another promising way to improve CZTS material properties. In theory thousands of different semiconductor materials with $I_2$-II-IV-$VI_4$ structures can be fabricated [58,59]. However, due to their similar electronic configurations, substitutes in the same periodic table groups as the CZTS components have been studied more extensively than other candidates. One interesting substitution is the replacement of Cu with Ag, resulting in $(Cu_{1-x}Ag_x)_2ZnSn(S,Se)_4$ or ACZTSSe. Due to the large size difference between Ag and Zn, Yuan et al. calculated that the substitution results in a reduced concentration of anti-site defects, even showing weak n-type conductivity [60], which was later confirmed by Chargarov et al., both in simulation and experimental work [61]. Gershon et al. developed a solar cell based on n-type AZTSe in a novel FTO/AZTSe/MoO$_3$/ITO structure, resulting in 5.2% power conversion efficiency (PCE) [62]. Guchhait et al. partially substituted Cu atoms with Ag by depositing ACZTS films on top of CZTS,

forming a Mo/CZTS/CdS/ITO structure, which showed an efficiency improvement from 4.9% to 7.2% [63]. A heterojunction between n-type AZTS and p-type CZTS resulted in 4.51% efficiency [64], while homojunction between n-type AZTS and p-type AZTS showed 0.87% PCE [65]. Structural studies showed that AZTS films are composed of larger grains and bandgap with higher mobility and lower charge-carrier concentration [66,67]. Replacing Zn with Cd is another highly studied substitution because of the close electronic structures of these atoms. Cd substitution can successfully tune the bandgap from 1.55 eV to 1.09 eV [68]. Su et al. fabricated a solar cell with 9.2% PCE by partially substituting Zn with Cd and observed an increase in grain size and a reduction in the ZnS secondary phase as well as a change in the depletion width, charge density and sheet resistance [69]. They also observed a phase transformation from kesterite to stannite for Cd/(Cd+Zn)>0.6. In addition to Cd, Zn can be replaced by a variety of other atoms, keeping the $Cu_2XSnS_4$ structure while X becomes Fe, Mn, or Mg [70,71]. Another interesting substitution is the replacement of Sn with Ge. Initial substitution was done by Hages et al., and with Ge/(Ge+Sn) = 0.3 composition the efficiency improved from 8.4% to 9.4% [72]. Later, Collard and Hillhouse studied the composition range of Ge/(Ge+Sn) from 0 to 0.9 and found that up to 50% Ge substitution increased the bandgap without any detrimental effects on optical properties, while also observing the highest efficiency of 11% at 25% substitution [73]. Kim et al. demonstrated an impressive 12.3% efficiency through Ge substitution [74] while pure $Cu_2ZnGeSe_4$ films showed 7.6% efficiency [75].

Except for the CZTS absorber, the interfaces between the CZTS/buffer layer and the CZTS/back contact also play important roles in achieving high efficiency CZTS solar cells. Gong et al. has published an extensive review which systematically identified major issues that limit the efficiency of CZTS solar cells from the aspects of the bulk of the absorber, grain boundaries of the absorber, the absorber/buffer interface and the absorber/back contact interface [24]. In their paper, they also proposed the potential improvement approaches and provided guidelines indicating where research efforts should be focused to achieve high efficiency CZTS solar cells.

## 4. CZTS Charge-Transport Layers in Perovskite Solar Cells

Since their introduction in 2009, the performance of perovskite solar cells (PSCs) has been improved remarkably. PSCs are regarded as third generation solar cells with a device structure similar to dye-sensitized solar cells. These devices consist of a bulk perovskite layer in the middle, where the light absorption occurs, and charge-separation is achieved by electron and hole transport layers. Therefore, developing new and effective charge-transport layers is an integral part of the perovskite research. To this day, many organic hole transport layers (HTLs) have been explored, such as 2,2',7,7'-Tetrakis[N,N-di(4-methoxyphenyl)amino]-9,9'-spirobifluorene (spiro-OMeTAD). However, an effective, low-cost hole transport layer with high stability is still missing. One promising alternative HTL material is CZTS, which has natural advantages including p-type conductivity and a composition of abundant and non-toxic components, as well as having many options to tune its bandgap. Recently, there have been an increasing number of studies using CZTS HTLs in PSCs, which have been summarized in this section and in Table 1.

In 2015, Wu et al. developed a $CH_3NH_3PbI_3$ perovskite solar cell with CZTS HTLs for the first time [76]. The device was fabricated in n-i-p architecture by spin coating CZTS nanoparticles (NPs) onto the perovskite layer, followed by a heat treatment at 100 °C for 10 min. Because HTL is deposited on top of perovskite in n-i-p architecture, there are serious constraints in deposition conditions, forcing researchers to use lower temperatures. In this study, the crystalline CZTS layer is achieved at low temperatures by using CZTS NPs produced by the hot injection method. Wu et al. also investigated the effects of particle size. Changing the reaction time resulted different size particles ranging from 8 ± 1 nm to 20 ± 2 nm. After optimizing the conditions, 200 nm thick CZTS HTLs deposited from 20 ± 2 nm particles showed the best performance of 12.75% efficiency, which was comparable to the 13.23% efficiency of the reference PSC with a standard spiro-OMeTAD HTL. CZTS HTLs showed similar open-circuit voltage ($V_{oc}$) and higher short-circuit current ($J_{sc}$) while having

lower fill factor (FF). The lower FF is associated with an increase in series resistance, which might be caused by the poor surface coverage of CZTS NPs.

**Table 1.** Performance of Perovskite solar cells with CZTS hole transport layers.

| Perovskite | Arch. | HTL | Efficiency | Year | Study Type | Ref. |
|---|---|---|---|---|---|---|
| $MAPbI_3$ | n-i-p | CZTS NPs | 12.75% | 2015 | Experimental | [76] |
| $MAPbI_3$ | p-i-n | CZTS NPs | 15.4% | 2016 | Experimental | [77] |
| $MAPbI_3$ | n-i-p | CZTSSe NPs | 10.72% | 2016 | Experimental | [78] |
| $MAPbI_3$ | n-i-p | RGO/CZTSSe | 10.08% | 2018 | Experimental | [79] |
| $MAPb_{1-x}Sn_xI_{3-y}Cl_y$ | n-i-p | CZTS NPs | 9.66% | 2018 | Experimental | [80] |
| $MAPbI_3$ | p-i-n | CZTS NPs | 6.02% | 2019 | Experimental | [81] |
| $CsPbBr_3$ | n-i-p | CZTS NPs | 4.84% | 2019 | Experimental | [82] |
| $MASnI_3$ | n-i-p | CZTSSe | 19.52% | 2019 | Simulation | [83] |
| $MAPbI_3$ | p-i-n | CZTSSe | 16.75% | 2019 | Simulation | [84] |
| $MAPbI_3$ | n-i-p | $Cu_2MnSnS_4$ | 8.35% | 2019 | Experimental | [85] |
| | | $Cu_2ZnSnS_4$ | 6.24% | | | |
| | | $Cu_2CoSnS_4$ | 7.55% | | | |
| | | $Cu_2NiSnS_4$ | 4.16% | | | |
| $MAPbI_3$ | n-i-p | $Cu_2CoSnS_4$ | 7.95% | 2020 | Experimental | [86] |
| | | $Cu_2NiSnS_4$ | 9.94% | | | |
| | | $Cu_2ZnSnS_4$ | 11.17% | | | |
| $MAPbI_3$ | n-i-p | Carbon/CZTS | 12.53% | 2020 | Experimental | [87] |

The positions of valence and conduction bands are important factors that affect recombination losses and charge collection. The general strategy to change the bandgap of CZTS is the replacement of S atoms with Se atoms in the crystal lattice, which causes an increase in valence band energy while reducing conduction band energy. Yuan et al. studied the effects of bandgap grading by replacing S with Se in an n-i-p $CH_3NH_3PbI_3$ PSC [78]. Both CZTS and CZTSe NPs were produced by the hot injection method. The bandgap is decreased from 1.64 eV for sulfide to 1.14 eV for selenide NPs. Selenium replacement caused a massive decrease in resistivity from 112.3 Ω·cm to 4.56 Ω·cm. Surface photovoltage spectroscopy showed better hole extraction in CZTSe HTL due to a larger valence band offset, as seen in Figure 6. However, electrochemical impedance spectroscopy showed that the CZTSe/$CH_3NH_3PbI_3$ interface has lower recombination resistance, resulting in much lower $V_{oc}$, which causes an efficiency drop from 10.72% (with CZTS HTL) to 9.72% (with CZTSe HTL).

Khanzada et al. investigated the effects of ligand engineering for CZTS NPs and tested their 'ligand free' NPs as HTL in a p-i-n PSC using $CH_3NH_3PbI_3$ [77]. Normally, long-chain ligands are used during the hot injection method to avoid large agglomerations and achieve small average particle sizes. However, these ligands limit particle-to-particle interaction in a film, reducing the charge-carrier movements. Moreover, ligands may cause formation of a carbon-based layer, further limiting the grain growth. In this study, Khanzada et al. developed the process given in Figure 7 to remove the long-chain oleylamine (OLA) ligands. They dissolved CZTS NPs in hexane and mixed it with an acetonitrile solution of $Me_3OBF_4$, which splits CZTS NPs and OLA ligands as a result of protonation or alkylation of ligands, leaving a positively surface charged CZTS NPs. Electrostatic stabilization can be achieved by weakly coordinating solvent molecules such as DMF. TEM results showed a reduction in particle size from 5.46 ± 1.1 nm to 5.15 ± 1.0 nm as well as a reduction in inter-particle space from 0.98 ± 0.26 nm to 0.54 ± 0.38 nm. The latter was seen as an indicator of ligand removal. Ligand free NPs showed three orders of magnitude higher hole mobility. Overall, the efficiency of PSC with CZTS HTL is increased from 12.2% to 15.4%.

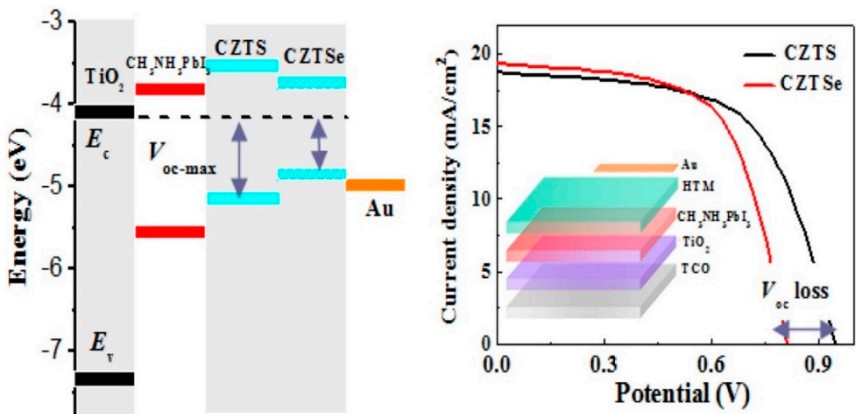

**Figure 6.** (**left**) Valance band maximum and conduction band minimum of perovskite solar cell (PSC) device components, (**right**) Device characteristics of PSCs with CZTS and CZTSe HTLs (reused from [78] with permission, licensed by Elsevier).

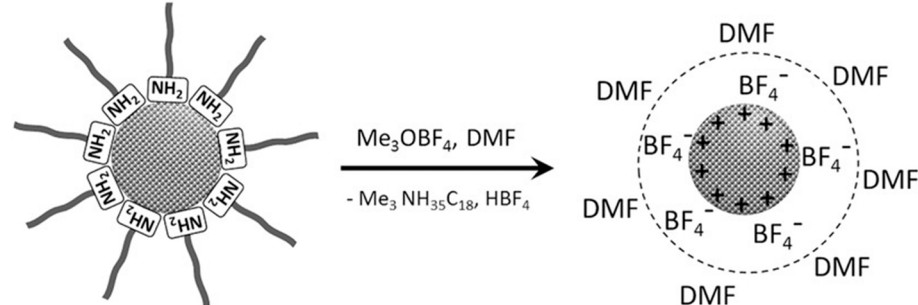

**Figure 7.** Schematic representation of oleylamine ligand removal from the CZTS NP surface (reused from [77] with permission, licensed by John Wiley and Sons).

In 2019, Zhou et al. fabricated a fully inorganic PSC with CZTS HTL for the first time [82]. CZTS NPs with $8 \pm 2$ nm sizes were produced by the hot injection method and deposited on top of the $CsPbBr_3$ layer to form a n-i-p PSC. Although $CsPbBr_3$ shows high stability compared to other inorganic perovskites, it is not widely used in solar cells as it has a bandgap of 2.3 eV, which is rather large for a single-junction solar cell. However, the valence bands of $CsPbBr_3$ and CZTS are fairly compatible (5.6 eV for CsPbBr3 and 5.4 eV for CZTS as seen in Figure 8), establishing a high efficiency charge-extraction. Overall, PSCs with CZTS HTL resulted in 4.84% efficiency, which is comparable with the 5.36% efficiency of the reference device with spiro-OMeTAD HTL.

Recently, new studies have explored co-using CZTS HTLs with carbon nanostructures. Nan et al. fabricated composite reduced graphene oxide and CZTSSe through an environmentally friendly sol-gel method [79]. PCSs with composite RGO/$Cu_2ZnSn(S_{0.5}Se_{0.5})_4$ HTLs showed 10.08% efficiency. The device also exhibited decent stability, with only a 14% performance loss after 500 h at room temperature operation. In another study, Cao et al. deposited a CZTS layer in-between carbon HTL and perovskite as a second HTL [87]. Devices with double CZTS/carbon HTL showed almost 50% better performances, reaching around 12.5% efficiency.

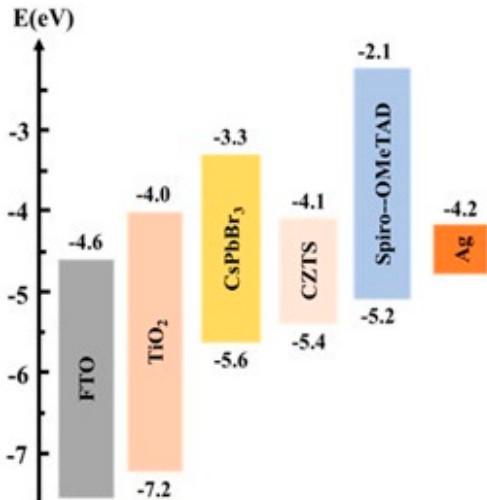

**Figure 8.** Energy level diagram of the CsPbBr$_3$ perovskite solar cells (reused from [82] with permission, licensed by John Wiley and Sons).

## 5. CZTS Photocathodes in Photoelectrochemical Water Splitting

Even if the efficiency of commercial solar cells has reached over 20%, because of the intermittent nature of solar energy, the electricity generated by solar panels at daytime or summer must be efficiently stored for peak electricity usage at night or in winter. Hydrogen (H$_2$) can be a promising way to store energy as a fuel. Although usage of H$_2$ is considered as clean, current H$_2$ production methods are mostly based on fossil fuels. However, water splitting offers a clean H$_2$ production method which can be coupled with solar energy in a variety of ways. Electricity generated by photovoltaics can be directly used for electrolysis of water to produce H$_2$, for which the efficiency is mainly dependent on the efficiency of the solar cells; therefore, it will not be detailed here. Compared to PV assisted electrolysis (PV-E), as shown in Figure 9a, a photochemical system is more cost effective. In a photochemical system (PC), as shown in Figure 9b, semiconductor powders are directly dispersed into electrolyte; after light excitation, the generated electrons and holes on the semiconductor surface can be used for the reduction/oxidation reaction. The efficiency of a PC system is normally one order lower than a PV-E system and has led to concerns about safety due to the existence of a mixture of O$_2$ and H$_2$ in the same chamber [88]. Photoelectrochemical (PEC), as shown in Figure 9c, water splitting, which is low-cost and can directly convert solar energy into clean fuels, will be discussed in detail in this paper. The free energy for splitting one molecule of H$_2$O into H$_2$ and O$_2$ under standard conditions is 237.2 kJ/mol, which is equal to 1.23 eV. Considering the energy loss during the electron transfer process at the semiconductor-liquid interface, the energy required for water splitting at a photoelectrode is normally circa 1.6–2.4 eV. Ideally, a single semiconductor material with a large energy bandgap (E$_g$) and a conduction band (CB) and valence band (VB) that straddles the electrochemical potentials of E$°$ (H$^+$/H$_2$) and E$°$ (O$_2$/H$_2$O), as shown in Figure 9d, can automatically drive water splitting under illumination. Therefore, early work on PEC photoelectrodes focused on wide bandgap metal oxides, which can absorb high energy Ultra-UV light from the sun and split water into H$_2$ and O$_2$. However, the large band gap of metal oxides leads to poor absorption for visible light, which limits the PEC efficiency. Because of its earth-abundant and non-toxic advantages, together with good optoelectronic properties such as a high light absorption coefficient, optimal bandgap and good stability, CZTS has been studied as a photocathode in PEC water splitting.

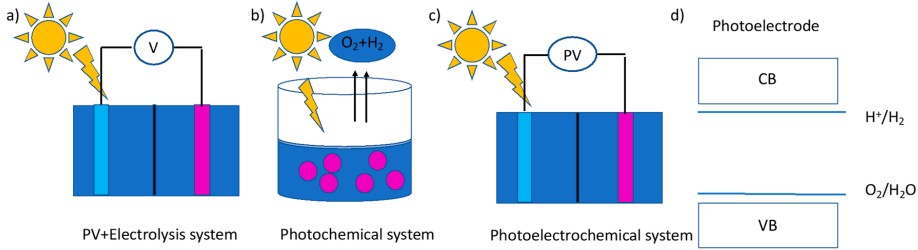

**Figure 9.** Scheme of (**a**) photovoltaic assisted electrolysis (PV-E) system, (**b**) photochemical (PC) system, (**c**) Photoelectrochemical (PEC) system and (**d**) the energy diagram in a single material PEC system.

Recent important studies that employs CZTS photocathodes have been summarized in Table 2. In 2010, Yokoyama et al. first demonstrated the application of CZTS as a photocathode. In their work, a series of surface modification works was done to improve the performance of CZTS photocathode. Firstly, in order to increase the charge separation, CdS and $TiO_2$ were deposited on top of CZTS thin-films for surface modification to accelerate charge transfer [89]. N-type CdS can form heterojunction with CZTS, and $TiO_2$ can protect the lower layer from corrosion in acid/alkaline solution. Secondly, in order to promote the $H_2$ evolution reaction, Platinum (Pt) was deposited as a catalyst for the CZTS photocathode. A type-II band structure was formed in the $CZTS/CdS/TiO_2/Pt$ photoathode, so the photogenerated electrons facilely transferred from the CZTS to the CdS, and then, through $TiO_2$, migrated into the Pt counter electrode where they were consumed to successfully generate $H_2$ gases. After optimization of the PEC system in electrolyte of different pH values, CZTS photocathode shows the best $H_2$ production to solar energy efficiency of around 1.2% and the highest photocurrent density of around 9 mA/cm$^2$. Yang et al. improved the photocurrent of the device with a similar structure to 13 mA/cm$^2$ through improving the quality of the CZTS layer [90]. In their work, the preparation procedure of the precursor solution containing the metal salts and thiourea was carefully controlled through the addition sequence of each chemical with the characterization of the Liquid Raman Spectra. Ros et al. further improved the solar energy to $H_2$ generation efficiency to 7% by optimizing the thickness of the atomic layer deposited (ALD) $TiO_2$ protecting layer and proceeded the test under highly acidic conditions (PH < 1) [91]. The ALD deposited $TiO_2$ layer plays the role of both an intermediate for electron transfer and a protection layer for the photocathode. A schematic of the system is given in Figure 10b. As shown in the J-V curve in Figure 10a, the Pt catalyst accelerates the $H_2$ reduction kinetics and increases the photocurrent at the photocathode. Due to the toxicity of Cd and the instability problem of CdS, Jiang et al. employed a novel $In_2S_3/CdS$ double layer on top of the CZTS film [92]. Combined with a Pt catalyst, the designed photocathode showed a significant improvement of stability during photoirradiation for PEC water splitting. The CZTS-based photocathode with novel surface modification achieved a half-cell solar-to-hydrogen efficiency (HC-STH) of 1.63%. The authors further demonstrated a bias-free PEC water splitting system by combining the above CZTS cathodes with a $BiVO_4$ counter photocathode. This was the first published bias free system based on a CZTS photocathode; the full cell power conversion efficiency (PCE) in the designed system was estimated to be 0.28%.

**Table 2.** Performance of a CZTS photoelectrode in PEC water splitting (under AM1.5 simulated sunlight without specific illustration).

| Photoelectrode | Preparation | Test Condition | PEC Performance | Ref. |
|---|---|---|---|---|
| $Pt/TiO_2/CdS/CZTS$ | Sputtering | 0.1M $Na_2SO_4$ (pH adjusted to 4.5 or 9.5) | HC-STC of 1.2%. | [89] |
| $Pt/s\text{-}TiO_2/CdS/CZTS$ | Spin coating | Phosphate-buffered aqueous solution (pH = 6.85) | 13 mA/cm$^2$ at −0.2 V vs reversable hydrogen electrode (RHE) | [90] |
| $Pt/TiO_2/i\text{-}ZnO{:}ITO/CdS/CZTS$ | Sputtering | 0.5 M $H_2SO_4$ (pH = 0.3) | HC-STC of 7% | [91] |
| $Pt/In_2S_3/CdS/Cu_2ZnSnS_4$ | Electrodeposition | Phosphate buffer solution (pH = 6.5) | HC-STC of 1.6% | [92] |
| $Pt/TiMo//CdS/Cd\text{-}CZTS$ | Spin coating | 1 M $K_2HPO_4/KH_2PO_4$ solution (pH = 7) | 17 mA /cm$^2$ at 0 V vs RHE | [93] |

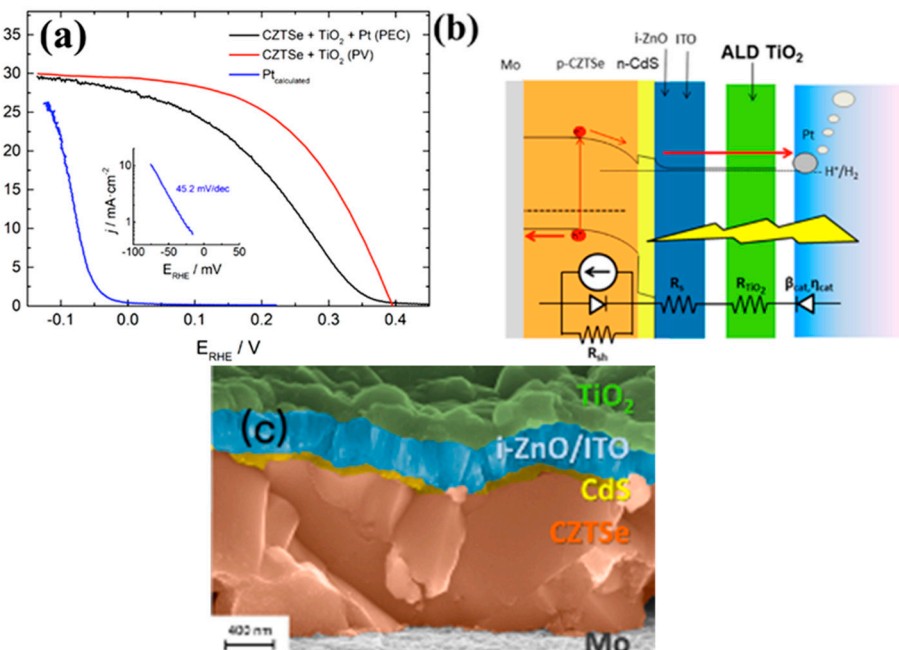

**Figure 10.** (**a**) J-V curve, (**b**) energy diagram, and (**c**) cross section SEM of TiO2 / i-ZnO:ITO / CdS / CZTS / Mo photocathode (reused from [91] with permission, licensed by American Chemical Society).

Adjusting the bandgap and CB/VB position is another efficient way to improve the PEC performance of the CZTS photocathode. As we have discussed previously, in the solar cell part, cation doping in CZTS could lead to changes in the bandgap, grain sizes, carrier concentration and defect concentration of CZTS. For example, Cd doping of Zn led to an increased grain size and suppressed formation of the ZnS secondary phase, which resulted in an improved solar cell efficiency. Tay et al. designed a photocathode based on a solution-processed $Cu_2Cd_{0.4}Zn_{0.6}SnS_4$ (CCZTS) photoabsorber [93]. After being coated with CdS/TiMo/Pt, the $Cu_2Cd_{0.4}Zn_{0.6}SnS_4$ photocathode yielded a photocurrent of 17 mA/cm$^2$ at 0 Vvs RHE, which is more than three times higher than pristine $Cu_2ZnSnS_4$. Hall measurement results revealed that CZTS have reduced carrier concentration and much higher carrier mobility after Cd doping, which could be caused by the reduction of $Cu_{Zn}$ anti-sites. The increased carrier mobility and lower carrier concentration leads to a wider space charge region and results in better charge collection. To further elucidate the reason for the lower onset potential and higher photocurrent for the CCZTS photocathode, an energy diagram based on X-ray photoelectron spectroscopy (XPS) and ultraviolet photoelectron spectroscopy (UPS) was studied. Because the Cd$^{2+}$ ion is larger than Zn$^{2+}$, substitution of Zn by Cd induces strain in the kesterite lattice of CZTS, which leads to a 'spike-like' conduction band offset at the CCZTS/CdS interface. The formation of a 'spike-like' interface plays a key role in reducing interface recombination.

A nanostructured CZTS photocathode has also been studied to enlarge the surface area of the Schottky junction that forms at the solid–electrolyte interface and so increase the separation of photo-generated carriers. In addition, the stronger light absorption and continuous shorter transport distance of photogenerated electrons in a nanostructured thin-film can also contribute to a higher photoelectrode performance than a dense sample. Wen et al. prepared porous nanocrystalline CZTS films by a facile metal organic decomposition (MOD) method through varying the amount of thiourea added in the precursor solution [94]. Compared with dense CZTS electrodes of similar crystallinity, composition and thickness, the porous CZTS photocathode yields much higher (three times) photocurrent. Results from capacitive currents under different scan rates shows that the porous CZTS film has a surface electrochemical area about four times larger than the dense film, which leads to a shorter transport distance of minority carriers and enhanced collected photocurrent in the porous structured CZTS. Instead of producing nanostructure in CZTS, Choi et al. applied CZTS

onto a three-dimensional (3D) $Cu_2ZnSnS_4$ (CZTS)/CdS/ZnO@steel composite photoanode [95]. The 3D structural photoanode was fabricated by solution method on FTO/steel mesh. A ZnO nanorod was grown by the hydrothermal method onto substrate, followed by CdS deposition using successive ionic layer adsorption reaction (SILAR) and then the spin coating of a CZTS layer using the wet chemical method. A coating of CZTS in this work can act as an efficient light sensitizer to enhance the visible absorption of the composite photoelectrode. A cascade type-II band alignment in the estimated energy diagram of the ZnO/CdS/CZTS system enables efficient charge separation and efficient electron transport across the interfaces of ZnO/CdS and CdS/CZTS. The combination effect of the efficient visible light absorption of CZTS, the cascade bandgaps diagram of the CZTS/CdS/ZnO nanorods and effective light absorption through the 3D structure together lead to a high photocurrent density of 12.5 mA/cm$^2$ at the 0.0 Vvs saturated calomel electrode (SCE).

## 6. Other Applications of CZTS

Thus far, we have summarized the mainstream application of CZTS in thin-film solar cells as well as two other promising applications as a charge-selective material and a photocathode in photoelectrochemical water splitting. However, there are other interesting applications where CZTS has been employed. In this section, wider potential applications are briefly summarized.

Thermoelectric devices are up-and-coming renewable energy technologies with huge potential that aim to convert heat waste into more useful electrical energy. CZTS can potentially be a thermoelectric material because of its highly distorted crystal structure, which results low thermal conductivity while having tuneable electrical properties. An important study showing that CZTS can exhibit good thermoelectrical properties was performed by Shi et al. In 2009 [96]. In that study, low thermal conductivity of CZTSe was combined with high electrical conductivity achieved by introducing indium impurities into a $Cu_2ZnSnSe_4$ lattice. An ideal composition was found at $Cu_2ZnSn_{0.9}In_{0.1}Se_4$, where the dimensionless figure of merit ZT reaches 0.37 at 700 K. Other studies employed Cu doping to achieve higher ZT values. Chen et al. reached ZT values of 0.7 at 450 °C thorough Cu doping [97]. Studies show that the thermoelectric properties of CZTSe increase with increasing temperatures [98], while stability issues become a problem at temperatures above 550 °C.

CZTS thin-films have been employed as photodetectors in a few studies [99–101], thanks to its appropriate optical properties. Singh et al. observed a significant improvement in the response rise time and photocurrent across near infrared to the visible region for Na doped CZTS photodetectors compared to un-doped CZTS [99]. Gour et al. fabricated rice-like nanostructured CZTS through sputtering in order to improve optical absorption, which resulted in a lower response rise time and decay times [100]. In a more recent study, Li et al. established a CZTS/ZnO heterojunction photodetector from vertically aligned ZnO nanorods and CZTS nanoparticles. The resulting photodetector exhibited high responsivity with fast rise and decay times [101].

CZTS can also be employed in a variety of sensors. Previously, CZTS-based liquefied petroleum gas (LPG) sensors with different heterojunction partners, such as polyaniline [102,103] and ZnO [104], have been presented. Studies have shown that the J-V characteristics of CZTS respond to LPG significantly differently than $N_2$, $O_2$, $H_2$ and $CO_2$, allowing detection. With its fast response time and high stability, CZTS could be a contestant to existing technologies in this area. Apart from LPG sensors, a novel biosensor for uric acid detection has been developed by Jain et al. based on uricase/CZTS/ITO/glass electrodes, which showed high sensitivity with a fast response time and a low detection limit towards uric acid [105].

## 7. Summary and Outlook

Kesterite CZTS, and its variants, has attracted great attention as a new absorber material for thin-film photovoltaic because of its earth-abundant, non-toxic and highly stable nature and the fact that it has similar properties to already commercial thin-film photovoltaic material CIGS. This makes it easier to develop higher performance CZTS-based devices by adapting concepts and know-how from

CIGS. Since its introduction, the record device efficiency of CZTS increased steadily until it reached a stagnation point a few years ago. Despite this slowdown, important developments and progress have been made in the past few years in terms of understanding the material properties as well as developing new strategies to reach higher performances. The main problems that limit the efficiency of CZTS solar cells, including the $V_{oc}$ deficit, non-ideal junction interfaces and deep-level intrinsic defects in the bulk CZTS, are yet to be addressed and solved. One proposed method to overcome these challenges is alkali metal doping. Specifically, Li, Na and K doping have been proven by many researchers to be highly beneficial for device performance. Cation substitution and alloying also promise interesting options. As a fairly complicated material, the components of CZTS can be substituted by keeping an $I_2$-II-IV-VI$_4$ structure, which allows the fine tuning of optoelectrical properties. Current promising alternatives are Ag substitution for Cu, Cd substitution for Zn and Ge substitution for Sn. However, there are also many more that could be explored. On the other hand, the advantages and interesting properties of CZTS also make it viable for various other applications. One of these potential applications is the use of CZTS as a hole transport layer in perovskite solar cells. A handful of studies have already proven it as a concept, and in the near future, by finely tuning the material properties, high performance perovskite devices could be produced. Another alternative application is using CZTS as a photocathode in photoelectrochemical water splitting for green hydrogen production. CZTS could have a bright future in this area, specifically as a result of its highly tuneable valence and conduction band edges through cation/anion substitution. CZTS can also be used as a thermoelectric material. Its highly disorganized crystal lattice causes low heat conductivity while its electronic properties can be improved by extrinsic doping. Lastly, some studies have shown that CZTS can also be used in photodetectors and gas sensors.

**Author Contributions:** Writing—original draft preparation, review and editing A.S.N. and M.W.; Writing—review and editing K.L.C.; Supervision K.L.C. All authors have read and agreed to the published version of the manuscript

**Funding:** The support of the Innovate UK-High Prospect project (No. 102470) and the EU-SCALENANO Project (No. NMP4-LA-2011-284486) are gratefully acknowledged. A.S.N. would also like to thank the Republic of Turkey Ministry of National Education for sponsoring his PhD studies under the YLSY sponsorship scheme.

**Conflicts of Interest:** The authors declare no conflict of interest.

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
