# Peer review of "Recent Development in Earth-Abundant Kesterite Materials and Their Applications"

_sustainability, doi:10.3390/su12125138_

Round 1
Reviewer 1 Report
This paper is a review paper on CZTS. The authors report not only CZTS solar cells, but also PVK solar cells and water splitting using them. It feels very interesting and deserves acceptance.
In the material properties of the first half, please cite the following references and make some modifications.
Tang Jiao Huang, Xuesong Yin, Guojun Qi, and Hao Gong,
Phys. Status Solidi RRL 8, No. 9, 735–762 (2014) /
Author Response
Thank you for the comments and suggestions. We have added the recommended reference to the paragraph about defect chemistry in the material properties section. We also referenced the same paper in an added paragraph at the end of section 3.
Reviewer 2 Report
1) Figure 1 is not adaquate for technical review for CZTS. It could be described in the text. I suggest delete it.
2) CZTS is Cu-deficient an Zn rich. So Cu vacancy and Zn-in-Cu sites are expected. Can you find a LTPL figure in CZTS? It would be very helpful for readers to understand defect chemistry.
Author Response
Thank you for the comments and suggestions. We have made the following changes based on your input:
1) We have deleted Figure 1 and moved the reference from which the data for the annual PV production by technology was taken to the appropriate section of the text.
2) We have made changes and additions to the defect chemistry paragraph in the material properties section. An LTPL figure was added as Figure 2c and explained in the text.
Reviewer 3 Report
The revision of K.L.Choy and coll about development and applications of Kesterita present in a good manner the present applications of this interesting material made from earth-abundant materials. It is very interesting, very well written and it is timely because it covers one of the most novel research to develop new material for solar cells and therefore I recommend publication with only minor changes.
The references, being actual an appropriate, there are some lack of recent revision with similar subject as S. Giraldo in Adv. Mat.2019 or J.B. Carda in Mater Letter 2015. In my pdf there are many comments saying (see Error reference source not found ) that should be corrected, although seem to correspond with references with Doi numbers.

Author Response
Thank you for the comments and suggestions. We have made the following changes based on your input:
1) “Error reference source not found” was an unexpected error caused by cross-reference function of MS Word which was used to keep track of figure and table captions. We removed all the cross-references and added figure captions manually. This problem now should be fixed.
2) We have added the recommended reference "J.B. Carda in Mater Letter 2015" to the 4th paragraph in section 3. We had actually cited "S. Giraldo in Adv. Mat. 2019" in the original manuscript and used a figure from this paper with permission since it was a well representation of possible substitution for CZTS components.